# Application of the theory of regulatory fit to promote adherence to evidence-based breast cancer screening recommendations: experimental versus longitudinal evidence

Serena Petrocchi  ,[1] Ramona Ludolph,[1] Nanon H M Labrie,[2] Peter Schulz[1]

► Prepublication history and additional materials for this paper is available online. To view these files, please visit the journal online (http://dx.doi.org/10.1136/bmjopen-2020-037748).

¹Institute of Communication & Health, Università della Svizzera italiana, Lugano, Switzerland
²Athena Institute, Vrije Universiteit Amsterdam, Amsterdam, The Netherlands

**Correspondence to**
Dr Serena Petrocchi;
serena.petrocchi@usi.ch

## ABSTRACT

**Objectives** To reduce overtreatment caused by overuse of screening, it is advisable to reduce the demand for mammography screening outside the recommended guidelines among women who are not yet eligible for inclusion in systematic screening programmes. According to principles of regulatory fit theory, people make decisions motivated by either orientation to achieving and maximising gains or avoiding losses. A study developed in two phases investigated whether video messages, explaining the risks and benefits of mammography screening for those not yet eligible, are perceived as persuasive

**Design** Phase 1 was an experimental study in which women's motivation orientation was experimentally induced and then they were exposed to a matching video message about mammography screening. A control group received a neutral stimulus. Phase 2 introduced a longitudinal component to study 1, adding a condition in which the messages did not match with the group's motivation orientation. Participants' natural motivation orientation was measured through a validated questionnaire

**Participants** 360 women participated in phase 1 and another 292 in phase 2. Participants' age ranged from 30 to 45 years, and had no history of breast cancer or known BReast CAncer gene (BRCA) 1/2 mutation.

**Results** In phase 1, a match between participants' motivation orientation and message content decreased the intention to seek mammography screening outside the recommended guidelines. Phase 2, however, did not show such an effect. Fear of breast cancer and risk perception were significantly related to intention to seek mammography screening

**Conclusions** Public health researchers should consider reducing the impact of negative emotions (ie, fear of breast cancer) and risk perception when promoting adherence to evidence-based breast cancer screening recommendations.

## INTRODUCTION

Breast cancer is one of the most common forms of cancer in women worldwide and the principal cause of cancer-related death

## Strengths and limitations of this study

► An experimental study (phase 1) and an experimental study with a longitudinal component (phase 2) were implemented applying principles from the theory of regulatory fit.
► An individual's goal-pursuit orientation was induced in phase 1 through a priming technique, and measured through a validated questionnaire in phase 2.
► Messages were tailored to create a match (or not) between message content and the individual's goal-pursuit orientation.
► Limitations of the studies included dropout rates (phase 2) and selection bias (due to cancer fear).

in the female population.[1] To promote early diagnosis, many EU countries have introduced systematic breast cancer screening programmes.[2] However, the age threshold to start inviting women to screening is in dispute.[3–5] The balance between the benefits (ie, reducing breast cancer mortality) and the harm associated with mammography (ie, X-ray exposure, overdiagnosis and false positive results; see [4–8]) is less certain for women aged under 50. Technologies for breast cancer screening have been constantly evolving, affecting evidence quality and suggested recommendations.[9] For these reasons recommended age for starting screening have varied from 40[10] to 45[11 12] to 50 years.[13 14]

In the last years, there has been a vast amount of research on screen intention, including barriers, enablers and how to get women with characteristics matching with the recommended guidelines to adhere to the screening programmes,[15–17]. There was also a progressive shift from persuading women to undergo screening to increasing their informed decision making.[18] Targeted information programmes and invitation materials

encouraging women to learn about the screening procedures increased levels of knowledge and supported decision making about their participation.[19] [20] Web-based dynamic decision aids, including pros, cons, controversies and overdiagnosis–overtreatment issues, have been found to improve the quality of information without reducing the screening participation rate.[21]

Other research tested communication programmes intending to inform women approaching 70 years of age about the benefits and harms of continuing screening.[22] [23] Similarly, non-high-risk women below the recommended age threshold seek and receive mammography screenings outside the suggested guidelines in the USA,[24] [25] Switzerland,[26] [27] Germany[28] and The Netherlands.[29] Studies show that women tend to overestimate the mortality reduction determined by breast cancer screening,[30] [31] and that they have unrealistic expectations regarding screening as reducing the risk of breast cancer[32] Moreover, social pressure in favour of breast cancer screening may stimulate a sense of moral obligation to participate[33] [34] even among young women.

Given the above-mentioned considerations, women under the age threshold for systematic breast cancer screening may consider the recommendation to avoid screening as counterintuitive, although scientifically supported, because of social pressure and the belief that cancer screening can save lives. The present research aimed to promote adherence to evidence-based recommendations on breast cancer screening among young women by activating a motivation system, such as regulatory orientation.[35]

## THEORY OF REGULATORY FIT

According to a popular psychological theory proposed by,[36] people show one of two regulatory orientations, which determines how they pursue their goals. They either show a promotion-focused orientation, meaning they eagerly strive towards the realisation of desired outcomes, or they show a prevention-focused orientation, emphasising the prevention of errors and losses and making them safety driven.[36] [37] While every individual has a natural tendency to lean more towards one orientation than the other, thus making it a measurable trait,[38] the regulatory orientation can also be experimentally induced.[35] [36] [39]

If individuals adopt a behaviour or processes a message highlighting goal-pursuit strategies that match their regulatory orientation, they experience a phenomenon called 'regulatory fit'.[35] For example, if a person with a promotion orientation reads a message highlighting strategies to achieve gains, a fit condition occurs. The same applies to someone with a prevention orientation processing a message emphasising strategies to avoid losses. Such a fit or match causes an 'it just feels right' perception, increasing the perceived value of the behaviour.[40]

The application of regulatory fit in the area of health communication is beneficial across various health contexts and outcomes.[41] Regulatory fit has been

consistently found to influence outcomes such as evaluation, behaviour and behavioural intention.[42] Some authors[40] showed that this 'it-just-feels-right' experience is also transferred to the context of persuasion, with positive effect of regulatory fit on the perceived persuasiveness of a message. A study by Uskul et al[43] in the context of tobacco use prevention among adolescents is in line with this finding. The effects of regulatory fit have also been extensively studied in the context of disease prevention and health promotion.[44] [45] In particular, some authors[43] applied the principles of regulatory fit to inform people about the benefits of regular cancer screenings. A systematic review[41] finds that the use of the principle of regulatory fit has the potential to increase the effectiveness of health communication across a range of health contexts and outcomes, making it a promising tool for tackling the problem of unwarranted demand for mammography screening outside the recommended guidelines.

No previous studies have tested messages designed according to the assumptions of regulatory fit to influence the intention to not engage in disease detection screening. This would challenge the intuitive perception that breast cancer screening leads to a mortality reduction determined by breast cancer in women over the age of 50,[30] [31] and the unrealistic expectations regarding screening as reducing the risk of breast cancer.[32] The purpose of the present research was to test whether health messages framed to correspond with a woman's regulatory orientation are effective in reducing the intention to ask for breast cancer screening in non-high-risk women under the age of 45, according to the local mammography screening guidelines. The following hypotheses have been tested:

HP1: a fit between the message frame and the regulatory orientation would lead to an immediate reduction of the intention to ask for breast cancer screening, in non-high-risk women under the age threshold indicated by the local guidelines.

HP2: a fit between the message frame and the regulatory orientation would lead to a reduction of the intention to ask for breast cancer screening, stable over time.

To this end, a study has been developed organised in two distinct phases: Phase 1 was an experimental study testing HP1, while phase 2 added a longitudinal component and tested HP2.

## METHODS
### Participants
#### Phase 1

An a priori power analysis applying G*Power V.3.1.9.2,[46] estimated a sample of 249 participants (α=0.05, d=0.95, η²=0.05; see reference [41]). Participants living in the Italian speaking, Swiss canton of Ticino completed an online survey from June to September 2016. The research was repeatedly advertised on the Facebook page of the University. Exclusion criteria were: a personal history of breast cancer, BRCA mutations, insufficient fluency in

Italian. The survey required women to answer each question before progressing to the next screen; as such there were no missing data. Participants received a 10 CHF supermarket voucher for their participation in phase 1. Before starting the questionnaire, participants completed a written informed consent by clicking on the corresponding button (ie, 'yes, I want to participate'; 'no, I do not want to participate').

Five hundred women from 30 to 45 years started the survey: 121 (16%) initiated the pretest questionnaires but dropped out. Nineteen of the women were excluded from the final sample because they did not complete the experimental manipulation. No differences emerged in the pretest variables between those who filled in only the pretest (N=140) and who filled in the entire survey (N=360). Participants were randomly assigned to prevention fit, promotion fit and control condition (see table 1). No differences were found between the intervention groups and the control group on sociodemographic variables.

### Phase 2

A priori power analysis estimated a sample size of 312. Recruitment took place from June to October 2017. The research was advertised through the Facebook page of the University and by RCSMedia Group, an Italy-based publishing group that uses participant panels. Inclusion/exclusion criteria were as for phase 1, with the addition that participants included in phase 1 could not participate in phase 2. Participants completed a written informed consent as for phase 1, and at the end received a 10 Swiss Francs /Euros supermarket voucher.

A total of 973 women aged from 30 to 45 filled in the pretest questionnaires (ie, pretest sample). Completed questionnaires (ie, analytical sample) were returned by 292 women with an attrition rate of 70%. Comparisons between the pretest sample and the analytical sample did not yield significant differences. A total of 292 women participated in the research (see table 1). This time, women aged 30–45 living in Ticino and Italy participated. Italian and Ticino-Swiss participants are not only comparable from a cultural and linguistic point of view, but also screening guidelines in Ticino and Italy are alike, inviting 50–74 yeas old biennially for mammography screening. No differences were found among the five groups regarding sociodemographic variables or other pretest variables.

### Process, measures and data collection
#### Phase 1
A pretest and post-test design with two experimental conditions and a control group was applied (see figure 1 for full details).

At pretest, the survey included measures of health status and health behaviours, a set of questions on past diagnosis of breast cancer, mammography, biopsy and knowledge of the Ticino screening programme. Women were rated on their fear of breast cancer, level of involvement in breast cancer and confidence in the benefit of mammography (see online supplemental material).

Participants were randomly assigned into a promotion fit, prevention fit or control condition. Regulatory priming manipulation was then induced following,[47] procedure (see online supplemental material). In the fit conditions, immediately after priming, the participants watched a video-message highlighting goal-pursuit strategies matching with the primed focus (see online supplemental material). The control group received a leaflet without any prompt for the regulatory orientation (see online supplemental material). In a pilot study, 30 women assessed the survey as clear and understandable.

### Phase 2
A pretest and post-test longitudinal design was applied with four experimental conditions, two fit conditions (promotion and prevention), two non-fit conditions (promotion and prevention) and a control group (see figure 2).

In the pretest (T0), participants replied to the same questions as for phase 1 (see online supplemental material). In phase 2, the regulatory focus orientation was measured with a questionnaire (see online supplemental material), rather than induced as in phase 1, because working with the trait regulatory focus would be more stable than a primed focus in a longitudinal design. Women were then identified according to their goal-pursuit main orientation (prevention orientation vs promotion orientation). Subsequently, participants were randomly assigned to the fit or non-fit condition or control group. In other words, randomisation was performed separately for prevention-oriented women and promotion-oriented women to ensure a balanced representation of orientations between the match and non-match conditions. Participants in the fit conditions watched two videos (at T1 and T2) emphasising the fit concerns (see online supplemental material). In the non-fit conditions, participants watched two videos (at T1 and T2) emphasising the non-fit concerns (see online supplemental material). In the control group, participants watched two videos (at T1 and T2) treating the topic of breast cancer prevention, but without any regulatory prompt (see online supplemental material).

A post-test questionnaire evaluated the women's intention to ask for opportunistic screening (T3). Ten days elapsed between each experimental phase. Three women from the general population assessed the videos as comprehensible and clear. The final survey was tested by fifteen women aged 30–45, who assessed it as clear and comprehensible.

### Patient and public involvement
#### Phase 1 and phase 2
Results from previous studies involving participants from Switzerland informed the present research (see reference 27). Participants were not directly involved in the design, conduct, recruitment, reporting or dissemination of the study results. An expert panel, composed of two health

**Table 1** Demographics of phase 1 and phase 2

| | Phase 1 | | | Phase 2 | | | | |
|---|---|---|---|---|---|---|---|---|
| | Promotion fit (N=122) | Prevention fit (N=130) | Control group (N=108) | Promotion fit (N=58) | Promotion Non-fit (N=57) | Prevention fit (N=74) | Prevention Non-fit (N=74) | Control group (N=29) |
| Age (range 30–45): M and (SD) | 36.55 (4.42) | 38.07 (4.57) | 38.37 (4.79) | 38.1 (4.96) | 38.53 (4.7) | 38.31 (4.44) | 37.93 (4.41) | 37.02 (4.99) |
| **Marital status** | | | | | | | | |
| Married | 73 (59%) | 77 (60%) | 69 (64%) | 36 (62%) | 41 (72%) | 55 (74%) | 53 (72%) | 22 (76%) |
| Single | 38 (31%) | 38 (30%) | 26 (24%) | 20 (35%) | 12 (21%) | 14 (19%) | 17 (23%) | 6 (21%) |
| Divorced/ separated/ widowed | 11 (10%) | 15 (10%) | 13 (12%) | 2 (3%) | 4 (7%) | 5 (7%) | 4 (5%) | 1 (3%) |
| **Educational level** | | | | | | | | |
| Elementary/ junior school | 2 (2%) | 2 (2%) | 3 (3%) | 1 (2%) | – | 1 (1%) | 4 (5%) | – |
| High school | 44 (34%) | 56 (46%) | 58 (54%) | 18 (31%) | 24 (43%) | 40 (54%) | 28 (38%) | 8 (28%) |
| University or post university degree | 84 (64%) | 64 (52%) | 47 (43%) | 39 (66%) | 33 (57%) | 33 (45%) | 42 (57%) | 21 (72%) |
| **Occupation** | | | | | | | | |
| Employed | 102 (84%) | 107 (82%) | 74 (69%) | 48 (83%) | 50 (88%) | 57 (77%) | 67 (91%) | 29 (90%) |
| Homemaker | 11 (9%) | 14 (11%) | 22 (20%) | 4 (7%) | 3 (5%) | 7 (9%) | 6 (8%) | 1 (3%) |
| Unemployed | 8 (6%) | 7 (5%) | 10 (9%) | 4 (7%) | 4 (7%) | 8 (11%) | 1 (1%) | 2 (7%) |
| Student | 1 (1%) | 2 (2%) | 2 (2%) | 2 (3%) | – | 2 (3%) | – | – |
| **Nationality** | | | | | | | | |
| Swiss | 97 (80%) | 101 (78%) | 73 (68%) | 10 (17%) | 15 (26%) | 16 (18%) | 13 (18%) | – |
| Italian | 21 (17%) | 23 (18%) | 26 (24%) | 47 (81%) | 40 (70%) | 53 (71%) | 58 (78%) | 27 (93%) |
| Other | 4 (3%) | 6 (4%) | 9 (8%) | 1 (2%) | 2 (4%) | 5 (7%) | 3 (4%) | 4 (7%) |
| **Mother tongue** | | | | | | | | |
| Italian | 117 (96%) | 122 (94%) | 94 (87%) | 54 (93%) | 54 (93%) | 68 (92%) | 71 (96%) | 27 (93%) |
| Other | 5 (4%) | 8 (6%) | 14 (13%) | 4 (7%) | 4 (7%) | 6 (8%) | 3 (4%) | 2 (7%) |

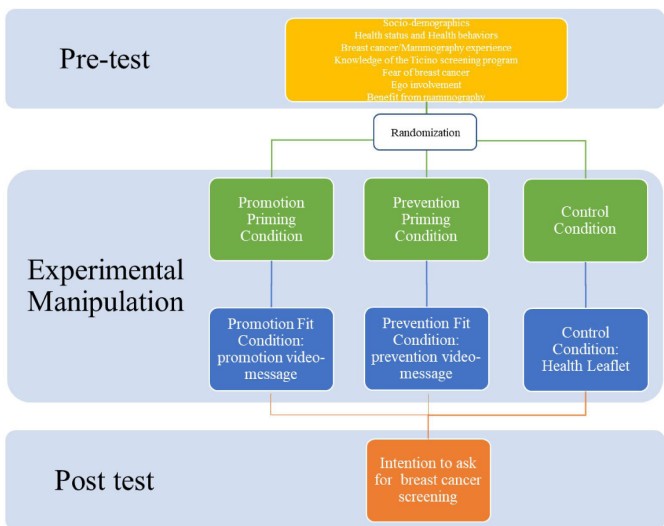

**Figure 1** Flow chart of the study 1.

communication professionals with expertise on regulatory fit theory, evaluated the message contents and the graphical aspects of the videos.

## Analytic strategy
### Phase 1 and phase 2
In both phase 1 and phase 2, data were normalised through reverse scoring and logarithmic transformations and there were no missing data.

In phase 1, an analysis of covariance (ANCOVA) tested the main hypothesis (HP1) of the study. The fit versus control conditions variable was inserted as independent variable. All the variables measured at the pretest were inserted as covariates. $\chi^2$ tests were conducted to evaluate whether the covariates might interact with the three experimental conditions in determining the intention to ask for breast cancer screening.

In phase 2, a repeated measure ANCOVA tested the main hypothesis (HP2) of the study. The fit versus non-fit versus control conditions variable was inserted as independent variable. All the variables measured at the pretest were inserted as covariates.

## RESULTS
### Phase 1
The ANCOVA analysis revealed that women in the two experimental conditions showed less intention to ask for breast cancer screening compared with the women in the control condition. Thus, when there is a fit between individual orientation (ie, a tendency to promote positive expected outcomes or to prevent negative outcomes for one's health) and the given message, then a persuasive effect is induced. There was no meaningful difference between the two manipulation conditions. Older women and women with higher levels of fear of breast cancer showed a greater intention to ask for breast cancer screening than younger ones and those with lower levels of fear. This evidence supports the assumption that regulatory orientation represents a motivational system able to overcome the impact of negative emotions and strengthen an individual's involvement in decision-making orientation. Descriptive data and results from the ANCOVA are displayed in table 2.

Further analyses were conducted to evaluate whether the covariates might interact with the three experimental conditions in determining the intention to ask for breast cancer screening. Analyses revealed only one association among the three groups of women and the past diagnoses of breast cancer among first degree-relatives, $\chi^2 (2)=12.98$, p=0.002. Women in the promotion fit condition had a lower number of breast cancer diagnoses among first-degree relatives than was expected ($z=-1.96$), while

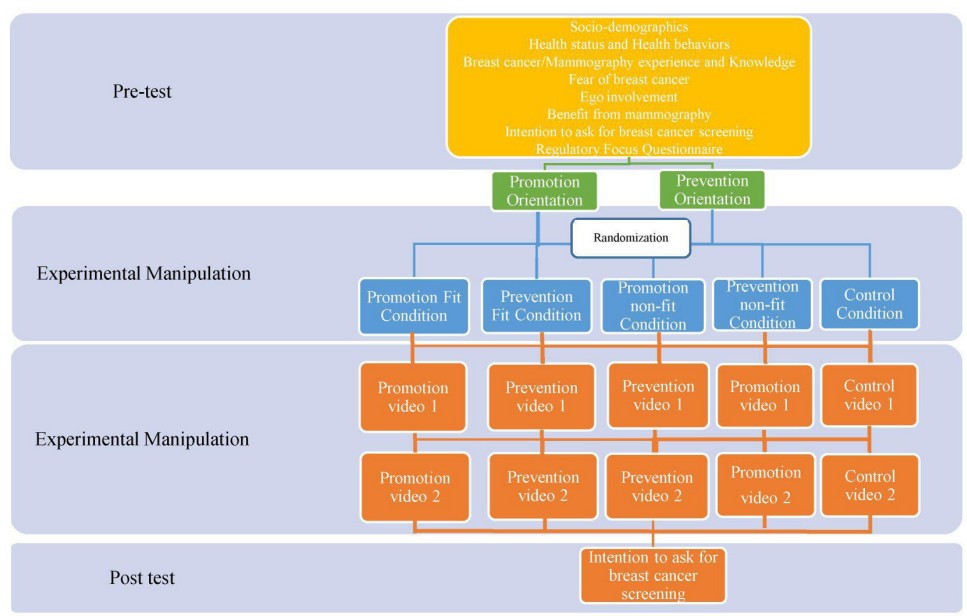

**Figure 2** Flow chart of the study 2.

**Table 2** Descriptive statistics of the pretest and post-test variables with frequencies (% frequencies between brackets) or means (SD between brackets), and results of the analyses

| | Phase 1 | | | Phase 2 | | | | |
|---|---|---|---|---|---|---|---|---|
| | Promotion fit | Prevention fit | Control group | Promotion fit | Promotion non-fit | Prevention fit | Prevention non-fit | Control group |
| **Pretest variables** | | | | | | | | |
| General health status | 3.88 (0.77) | 3.77 (0.87) | 3.7 (0.87) | 3.79 (0.79) | 3.63 (0.67) | 3.66 (0.76) | 3.70 (0.77) | 3.76 (0.69) |
| Physical activity | 2.45 (1.85) | 2.47 (1.69) | 2.43 (1.92) | 0.78 (0.42) | 0.81 (0.39) | 0.76 (0.46) | 0.72 (0.45) | 0.86 (0.35) |
| Smoking habits | 1.86 (4.85) | 1.99 (4.99) | 3.32 (6.42) | 3.53 (5.4) | 3.12 (4.66) | 4.93 (5.59) | 3.19 (5.15) | 7.22 (5.4) |
| Alcohol consumption | 1.92 (2.79) | 1.71 (2.27) | 1.42 (2.14) | 3 (2.26) | 2.66 (2.15) | 3.27 (4.13) | 2.67 (3.54) | 3.1 (4) |
| Fear of breast Cancer | 3.4 (.85) | 3.4 (.81) | 3.4 (1) | 3.75 (.95) | 3.59 (.91) | 3.79 (.95) | 3.83 (.93) | 3.68 (1) |
| Ego-involvement | 5.9 (1.1) | 5.8 (1.27) | 5.9 (1.3) | – | – | – | – | – |
| Benefit for mammography | 3.9 (0.62) | 3.8 (0.62) | 4 (0.74) | 4.1 (0.75) | 4.12 (0.73) | 4.16 (0.68) | 4.16 (0.65) | 3.94 (0.78) |
| Intention to ask for BC screening | – | – | – | 3.35 (1.33) | 3.35 (1.29) | 3.44 (1.22) | 3.31 (1.40) | 3.45 (1.41) |
| **Diet** | | | | | | | | |
| No | 46 (38%) | 49 (37%) | 39 (36%) | 24 (41%) | 30 (53%) | 29 (39%) | 27 (37%) | 13 (45%) |
| Yes | 76 (62%) | 81 (62%) | 69 (64%) | 34 (59%) | 27 (47%) | 45 (61%) | 47 (63%) | 16 (55%) |
| **BC among relatives** | | | | | | | | |
| No | 117 (96%) | 117 (90%) | 89 (82%) | 52 (90%) | 48 (84%) | 65 (88%) | 67 (90%) | 28 (97%) |
| Yes (mother) | 4 (3%) | 8 (6%) | 17 (16%) | 5 (9%) | 6 (11%) | 7 (10%) | 7 (10%) | 1 (3%) |
| Do not know | 1 (1%) | 5 (4%) | 2 (2%) | 1 (1%) | 3 (5%) | 2 (2%) | – | – |
| **Mammography** | | | | | | | | |
| No | 100 (82%) | 100 (77%) | 72 (67%) | 40 (69%) | 38 (67%) | 50 (68%) | 50 (68%) | 23 (79%) |
| Yes | 22 (18%) | 30 (23%) | 36 (33%) | 18 (31%) | 19 (33%) | 24 (32%) | 24 (32%) | 6 (21%) |
| **Biopsy** | | | | | | | | |
| No | 17 (77%) | 27 (90%) | 26 (72%) | 57 (98%) | 55 (97%) | 67 (91%) | 71 (96%) | 29 (100%) |
| Yes | 5 (23%) | 3 (10%) | 10 (28%) | 1 (2%) | 2 (3%) | 7 (9%) | 3 (4%) | – |
| **Knowledge of BC screening programme** | | | | | | | | |
| No | 76 (62%) | 69 (53%) | 64 (59%) | 21 (36%) | 23 (40%) | 26 (35%) | 27 (37%) | 7 (24%) |
| Yes | 46 (38%) | 61 (47%) | 44 (41%) | 37 (64%) | 34 (60%) | 48 (65%) | 47 (63%) | 22 (76%) |
| **Knowledge of the age thresholds for BC screening programme** | | | | | | | | |
| Do not know | 16 (35%) | 24 (39%) | 13 (30%) | 21 (36%) | 32 (56%) | 26 (35%) | 27 (37%) | 7 (24%) |
| Wrong | 22 (48%) | 28 (46%) | 30 (68%) | 29 (50%) | 18 (32%) | 34 (46%) | 37 (50%) | 18 (62%) |
| Correct | 8 (17%) | 9 (15%) | 1 (2%) | 8 (14%) | 7 (12%) | 14 (19%) | 10 (13%) | 14 (14%) |
| **Post-test variables** | | | | | | | | |
| Intention to ask for BC screening | 2.20 (1.05) | 2.26 (1.06) | 3.36 (1.33) | 3.02 (1.61) | 2.89 (1.48) | 3.17 (1.48) | 3 (1.54) | 2.78 (1.49) |
| Results from ANCOVA* or repeated measures ANCOVA† | $F^*$ (2, 319)=49.57, p<0.0001, $\eta^2_p$=0.24 | | | Within subject comparison between preintention and postintention: F† (1, 267.91)=5.10, p=0.025, partial $\eta^2$=0.02<br>Between subject comparisons among groups: F† (4, 284)=0.43, p>0.05 | | | | |
| | Promotion fit versus control condition<br>t(319)=−8.80, p<0.0001, r=0.44 | | | | | | | |
| | Prevention fit versus control condition<br>t(319)=−8.80, p<0.0001, r=0.44 | | | | | | | |
| | Significant covariates | | | | | | | |
| | Fear of BC: F(1, 319)=6.81, p=0.010, partial $\eta^2_p$=0.02 | | | Fear of BC: t(284) = 2.76, p=0.006, B=0.24, partial $\eta^2$=0.03 (95% low CI=0.07, 95% high CI=0.42) | | | | |
| | Age: F(1, 319)=26.20, p<0.0001, partial $\eta^2_p$=0.08 | | | Age, t(284) = 6.26, p<0.0001, B=0.11, partial $\eta^2$=0.12 (95% low CI=0.08, 95% high CI=0.15) | | | | |
| | | | | Risk perception, t(284) = 2.26, p=0.024, B=0.37, partial $\eta^2$=0.02 (95% low CI=0.05, 95% high CI=0.70). | | | | |

*Ancova
†Repeated Measures Ancova
ANCOVA, analysis of covariance; BC, breast cancer.

women in the control condition had a higher number than expected ($z$=2.8). The subsequent ANCOVA did not find any significant interaction between past diagnosis of breast cancer among first-degree relatives and the experimental manipulations, therefore, demonstrating that the regulatory fit genuinely influences the intention.

## Phase 2

There was a general significant decrease of the intention from pre-evaluation to postevaluation across groups, but no significant differences among them, indicating that the scores of the post-test intention among the five groups were in general the same. Among the covariates older women, greater fear of breast cancer and greater risk perception were associated with greater post-test intention compared with the opposite. Table 2 shows descriptive statistics and results from the analysis.

The intervention effect was not significant either when the two fit conditions and the two non-fit conditions were collapsed into two categories (ie, comparison among fit condition vs unfit condition vs control) as done in phase 1, even though a general decrease in the postintention across groups was found as before. Risk perception was tested as a moderator, but the analysis was not significant.

## General discussion

The present research shows inconsistent results. Phase 1 confirmed the hypothesised effect of the intervention on the intention to seek mammography screening before the age of 45, with a reduction of the intention when a fit between the message frame and the individual's regulatory focus occurred. Longitudinal results from phase 2 demonstrated that this effect was not significant over 1 month, although a general decrease of the intention across groups was observed. Even though further evidence is needed to confirm our results, it still seems that the 'just-feels-right' experience appears to be insufficient to convince non high-risk women under the age threshold to avoid systematic breast cancer screening outside of the recommended guidelines.

Our results could genuinely reflect the fact that the regulatory fit is not sufficient to induce a long-term decrease in women's intentions or could be an artefact of the research itself. Phase 1 and phase 2 applied two different ways to evoke a regulatory orientation. Phase 1 primed the individuals' regulatory orientation, whereas phase 2 measured it with a questionnaire to overcome a limitation of phase 1 and explore a different aspect of the theory. One could argue that the different ways to induce versus measure the regulatory orientation could have influenced the persuasiveness of the message and so its effectiveness. However, researchers of regulatory orientation suggest that there is no difference between the two procedures.[39] Therefore, we could exclude that the two methods have had a differential impact on post-test intention. Possible differences in the cultural milieu of Italian-speaking Swiss and Italian participants might make the population primed to receive or primed to ignore the intervention. However, to the best of our knowledge, there is no study comparing different cultural environments in the propensity to be primed or not.

The relatively small sample size and the recruitment strategies could have influenced the power of the analyses, the sample composition and, ultimately, the significance of the results. However, there is no such concern in phase 2 since the effect due to the intervention was not significant either when the two fit conditions and the two non-fit conditions were collapsed into two categories.

Finally, a variable might have moderated the association between intervention and intention. As[48] demonstrated, individuals' consideration of future consequences of a particular behaviour influences the effectiveness of framing techniques in predicting risk perceptions, attitudes and behavioural intentions regarding health-related advertisements. In our research, the risk perception was tested as a moderator variable, but the analyses yielded no significant results.

Fear of breast cancer, age, and risk perception (only in phase 2) were significantly related to women's intentions. The predicting role of age is not surprising because, approaching the age of 50, women are invited to undertake regular mammography screening in Ticino and in Italy. Risk perception and fear of breast cancer are the most sensitive variables. Breast cancer naturally evokes negative emotions.[27 49–51] Moreover, the benefits of mammography screening often seem to be overestimated.[30 31] Therefore, it is challenging to develop effective health messages promoting the adherence to breast cancer screening guidelines for young women based on factual information. As messages based on the principles of regulatory fit take the motivational orientations of recipients into account, they go beyond the effectiveness of purely providing information. Here, messages building on the theory of regulatory fit did not seem to offer a new way to overcome the 'emotional barrier' generated by the fear of breast cancer. However, phase 2 demonstrated a general 'pedagogical effect' deriving from talking about the topic of breast cancer screening without evoking a boomerang effect (ie, an increase of intention instead of a decrease).

The present research has several limitations. We experienced high dropout rates, especially in phase 2. The high drop-out rates may be related to the topic of breast cancer itself or the fear associated with it. One could assume that women with a low level of fear of breast cancer may have decided not to take part in our research, and this may have created a selection bias that could affect the generalisability of the results. A second limitation concerns the fact that we measured the intention to ask for breast cancer screening, not the actual behaviour. Although according to many theories in the field of health promotion (eg, Health Belief Model), the intention is a valid predictor of the actual behaviour, it would be beneficial if future research followed women until the moment they actually have a mammography.

In conclusion, it seems that by framing health messages that conform to a promotion or prevention focus, a decrease in the intention to ask for merely preventive opportunistic mammography screening is observed; but this takes place only immediately after message exposure. The influence decreases over time, and the messages lose their predictive

effects after 1 month. This may be because breast cancer fear/opinions are very deeply ingrained in women and one/two messages cannot change that. Accordingly, possibly results from phase 1 are valid, but repeated exposure to more than one regulatory fit message is needed to change viewpoints in the long term.

Even though our results only partially confirmed our hypothesis, there are substantial implications for future research. The results demonstrate that fear of breast cancer and risk perception are the main challenges to face in order to promote adherence to evidence-based recommendations on breast cancer screening. Public health researchers must investigate what factors may increase the effectiveness of health information. According to our evidence, future research may consider understanding how to reduce the impact of negative emotions rather than try to overcome their effect. For example, a research[52] found that humour in health messages reduces the anxiety associated with performing cancer screening. Humour may be implemented in health messages aimed to promote evidence-based breast cancer screening recommendations. Reducing the number of unnecessary breast cancer screenings would thus allow the prevention of avoidable false positive and false negative diagnoses and unjustifiable mental and physical suffering for women. In the long term, this would also enable policymakers and health professionals to allocate scarce resources for disease prevention, detection and treatment in a more effective way.

**Acknowledgements** We are very grateful to Professor John Hodgson for his thoughtful revisions.

**Contributors** SP and RL drafted the first version of the manuscript. All authors contributed to writing and critically revising it and approved its final version. PS acquired funding. NHML, PS and RL designed study 1 and prepared the materials. PS, SP and RL designed study 2 and prepared the materials. SP and RL collected data for study 1. SP collected data for study 2. SP performed the analyses for study 1 and study 2.

**Funding** This work was supported by the Swiss National Science Foundation (grant number FNS 100019-153131/1).

**Competing interests** None declared.

**Patient consent for publication** Not required.

**Ethics approval** The University's Ethical Committee approved phase 1 and phase 2.

**Provenance and peer review** Not commissioned; externally peer reviewed.

**Data availability statement** No data are available. Data are available on request.

**ORCID iD**
Serena Petrocchi http://orcid.org/0000-0002-7223-8240

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
