## [Reviewer comments · BMJ Open]

ARTICLE DETAILS

TITLE (PROVISIONAL)	An Application of the Theory of Regulatory Fit to Promote Adherence to Evidence-Based Breast Cancer Screening Recommendations: Experimental vs. Longitudinal Evidence
AUTHORS	Petrocchi, Serena; Ludolph, Ramona; Labrie, Nanon; Schulz, Peter

VERSION 1 – REVIEW

REVIEWER	Lisa Parker University of Sydney, Australia
REVIEW RETURNED	09-Mar-2020

GENERAL COMMENTS	Review for BMJ Open Manuscript 2020-037748 Synopsis This paper explores the effectiveness of different strategies for encouraging the population to adhere to breast screening guidelines, specifically in relation to reducing intention to screen at a younger age than recommended. Comments: Overall – this is an important topic and one that deserves research attention. My main concerns were that the paper was written up as two separate studies. I think more work could be done to write up in a more traditional style, with just one method, results, discussion section. Abstract: I found the use of terminology confusing here – many of your casual readers will not have heard about the principle of regulatory fit so I would encourage you to use more explanatory language in this section to maximise understanding. Introduction: There has been a vast amount of research on intention to screen, including barriers, enablers and how to get people to adhere to screening guidelines. Most of this is about encouraging people to attend screening, rather than to avoid unnecessary screening. I would have liked you to refer more to this literature and also to make more of an argument for your paper filling a gap. I found some of the terminology confusing. For example, the phrase ‘medically not indicated breast cancer screening’ is not intuitive. This may be a translation issue, and I wonder if you can find something that works better. You could try ‘screening that is not recommended by local guidelines’ or something similar. I also struggle with the comment about lay people’s “common sense” (p
---

	8, line 8-9). Perhaps either explain why you think it's common sense to attend screening more often than recommended, or use a different phrase. I struggled to understand your description of the studies on p 8, line 29-30. For example, even though you had briefly introduced the theory of regularly fit, I didn't know what you meant by 'comparing two fit conditions vs. two non-fit conditions'. The meaning is explained later, but it's confusing in this section, I'd encourage you to explain more, or to leave it out of the introduction and put all the details in the methods section. Methods: As mentioned, I'd suggest you combine the methods sections. Results: Some of the text here is quite dense with quantitative results and confidence intervals, making it very difficult to read. I'd suggest putting the quantitative detail in tables, and using the text just to highlight important findings, without repeating information that is in tables. Please note that I am not a quantitative researcher and cannot provide specialist statistical review. Discussion: I was interested in your inconsistent results between the two phases of the study. I wonder whether the nationality of the participants had an impact (study 1 was predominantly with Swiss nationals, study 2 was predominantly Italian nationals). Perhaps you could comment on this. For example, is there a different cultural milieu in the two groups that might make the population primed to receive or primed to ignore your intervention? Again, there was limited reference to the vast literature exploring barriers / enablers to adherence to breast cancer screening guidance. This might help to situate your study more clearly in the canon, assist with your interpretation and help to guide future research.
--	--

REVIEWER	Jolyn Hersch The University of Sydney, Australia
REVIEW RETURNED	30-Mar-2020

GENERAL COMMENTS	An Application of the Theory of Regulatory Fit to Promote Adherence to Evidence-Based Breast Cancer Screening Recommendations: Experimental vs. Longitudinal Evidence Journal: BMJ Open Manuscript: bmjopen-2020-037748 This paper reports two connected studies seeking to address an important public health objective, namely reducing inappropriate overuse of screening (in this case mammography among women under 50 years of age). This research uses psychological theory to generate and test hypotheses about enhancing communication around this topic by taking into account motivational factors that may interact with information provision. I think this work could ultimately make an interesting and valuable contribution to the literature, and I commend the authors for their efforts. However, the research is somewhat complex and I feel the manuscript
--

needs some refinement before it would be suitable for publication. Please see detailed comments below.

MAIN POINTS

1. Title: I find the title ambiguous with respect to the kind of 'adherence to recommendations' this paper addresses. The research purpose is articulated nicely at the top of p8: "to test whether health messages framed to correspond with a woman's regulatory focus orientation are effective in reducing the intention to ask for medically not indicated breast cancer screening under the age of 50". Could authors consider whether their title could be more informative by incorporating some of this wording (re reducing screening intentions in women under 50)?

2. Abstract: Could the authors consider including some wording (e.g. 'overuse' or a related term) in the abstract to facilitate connecting the paper with the broader literature around the topic?

3. I think table 2 should be converted into two flow charts to show more clearly the sequence of steps in each study, including randomisation to conditions, measurement of variables at time points, manipulation/priming of orientation, tailoring of messages to personal orientation etc.

4. As intention is the primary outcome, more detail is needed about its measurement (p11). What was the response scale/options and possible range of scores? This is crucial to interpret table 4.

5. Analytic strategy (study 1, p11) needs elaboration – for example 'We used analysis of covariance to...' Also, I notice that "There were not missing data." Is that because the survey required women to answer every question before progressing to the next screen? If so, please state this.

6. Results, opening sentence (study 1, p11): Please double-check the numbers – is it just coincidence that the numbers are exactly the same for both conditions?

7. Discussion (study 1, p12): Please include a sentence here to briefly recap the main finding/s.

8. "292 participants were randomly assigned to five conditions" (p13) – wouldn't it be accurate to say 973 women in the pre-test sample were randomised? Also, does that sentence contradict this? "Participants were randomly assigned to the fit or non-fit condition or the control group."

9. Procedure (study 2, p14): "...questionnaire measuring women's regulatory focus. The latest was applied because working with the trait regulatory focus would be more stable than a primed focus..." It seems to me this is an important point about a key difference between studies 1 and 2. I suggest explaining this in the discussion section of study 1 to help readers understand why study 2 was done (e.g. to address a limitation of study 1, explore a different aspect of theory).

10. I'm confused by these two sentences (p15) as they appear similar yet different: "The within-subject effect reached

	significance, $F(1, 268.25)=5.34$, $p=.021$, with a general decreasing of the intention from the pre-test ($M=3.38$, $SD=1.32$) to the post-test measurement ($M=3$, $SD=1.52$).” / “There was a significant within subjects effect, $F(1, 267.91)=5.10$, $p=.025$, partial $\eta^2=.02$, indicating that there was a general decrease of the intention from pre to post-evaluation...” 11. Several times throughout the manuscript the authors appear to cite their own previous work as ‘blind for review’. Perhaps this was necessary for previous submission to a different journal, but I don’t think it’s required for BMJ Open since peer review is fully open anyway. When submitting any revision of this manuscript please include full details of all references. MINOR POINTS 12. Introduction (p6): “possible harms associated with regular mammography below the age of 50 are recognized” – please name the main harms (i.e. overdiagnosis and false positives). 13. Introduction (p6): “many women below the age of 50 seek and receive screenings” – I haven’t checked the references but assume they cover data from various countries? I suggest saying so. 14. This is a matter of taste and the authors might disagree, but I suggest moving the sentences about recruitment and eligibility (bottom of p9) to earlier, under the heading Participants. (And likewise for study 2). Also, although the Facebook advertisements obviously were targeted to women in Ticino, they presumably appeared on some public Facebook pages – so was living in Ticino actually part of the eligibility criteria or could anyone visiting these pages take part? 15. I have never heard of the Personal Involvement Inventory (p10) – could the authors please add a sentence to briefly explain what it is and why it was included? 16. On p13 “Italian and Ticinese-Swiss participants ... follow the same rules for their breast cancer screening programs.” Please briefly state the program ‘rules’ referred to (here or in the intro). 17. I am not sure what you mean by “the ‘just-feels-right’ experience” (p16) – please explain. 18. Please briefly describe the process of obtaining informed consent from study participants.
--	--

REVIEWER	Stephen W. Duffy Queen Mary University of London, UK
REVIEW RETURNED	10-May-2020

GENERAL COMMENTS	1. This paper reports on a comparison of different communication strategies aimed at changing health behaviour with respect to mammography screening participation. The study design seems appropriate and the work has been conducted correctly, but two major problems with this paper occur to the reader. First, it seems to be addressing the wrong problem. In Europe and North America, the problem isn’t too many people seeking mammography screening: the problem is that large numbers of
---

	women who need mammography screening do not get it. In the UK, 30% of those invited to the national programme do not attend. The figure is higher in many other European countries. 2. A second issue is the assumption that screening in the general population in women aged under 50 is assumed to be detrimental. The paper justifies this by reference to a highly selective set of publications. There is evidence from both experimental (e.g., Nyström et al, J Med Screen 2017; 24: 34-42) and observational studies (e.g, Hellquist et al, Cancer 2011; 117: 714-22) that mammography screening in women aged 40-49 is effective in preventing breast cancer deaths. Some balance here would be welcome. 3. The reader is somewhat thrown in at the deep end with respect to the language of social psychology, from abstract onwards. I am not sure what could be done about this without unacceptable increases in word counts. However, I suggest that the authors try to signpost the concepts for the more general reader. 4. It is a limitation that post-intervention, only intention is reported, not actually screening participation. Is it possible to obtain data on actual participation post-intervention? If not, this should be acknowledged as a limitation.
--	---

VERSION 1 – AUTHOR RESPONSE

Reviewers' Comments to Author:

Reviewer: 1

Reviewer Name: Lisa Parker

Institution and Country: University of Sydney, Australia

Please state any competing interests or state 'None declared': none declared

Synopsis

This paper explores the effectiveness of different strategies for encouraging the population to adhere to breast screening guidelines, specifically in relation to reducing intention to screen at a younger age than recommended.

Comments:

Overall – this is an important topic and one that deserves research attention. My main concerns were that the paper was written up as two separate studies. I think more work could be done to write up in a more traditional style, with just one method, results, discussion section.

Authors: thank you for your appreciation and suggestions. We have revised the paper according to your comments and the other two reviewers' comments via track changes.

We carefully considered your suggestion to collapse the two studies into one (one method and one result section). However, we feel that the ms with one study, instead of two, would lack of clarity. Undoubtedly, there are similarities between study 1 and study 2, but at the same time, there are also many differences.

In study 1 we primed/induced the regulatory focus through a well-established technique (Higgins, 2001). The second study measured the regulatory focus through a questionnaire (which is different from the priming).

In the first study, we collected data with an experiment with randomization into 3 conditions. The second study added a longitudinal component with randomization into 5 conditions.

The second study is the consequence of the first one. The first study explore the topic, while the second study is meant to expand the results and to improve the methodology.

Finally results from the two studies are different and need to be kept separately.

For all the reasons mentioned above, we decided to keep the two studies separately. However, we understand your suggestion to give the reader a more comprehensive paper, avoiding repetitions and technicalities that makes the text difficult to read. For this reason, we worked a lot on the ms and implemented several changes:

- abstract: we have changed the terminology about the theory of regulatory fit that created confusion. We now give the reader the possibility to understand straightforward what we have done
- method sections: we deleted many parts to avoid repetition. However, we have created a supplementary material with full information for whom might be interested
- in the results section we avoid technicalities and put all the statistics into the existing tables
- We developed two flowcharts with a description of the method and phases of the two studies. You may found it in the tables file.

Abstract: I found the use of terminology confusing here – many of your casual readers will not have heard about the principle of regulatory fit so I would encourage you to use more explanatory language in this section to maximise understanding.

Authors: we have changed the terminology regarding the theory of regulatory fit and used explanatory language

Introduction:

There has been a vast amount of research on intention to screen, including barriers, enablers and how to get people to adhere to screening guidelines. Most of this is about encouraging people to attend screening, rather than to avoid unnecessary screening. I would have liked you to refer more to this literature and also to make more of an argument for your paper filling a gap.

Authors: thank you for your suggestion. The reason why women ask (or not) for screening is an interesting topic. Although a complete review of this literature is beyond the scope of our paper, we have added a sentence in the introduction discussing this point. We also added a couple of references (*There has been a vast amount of research investigating the intentions of women to adhere to screening guidelines and encouraging women with characteristics that match with the national guidelines to engage in screening*). We agree with you regarding the fact that most of this literature encourage women to attend screening, rather than avoiding unnecessary screening. We discussed this point in the first paragraph of the discussion section.

I found some of the terminology confusing. For example, the phrase 'medically not indicated breast cancer screening' is not intuitive. This may be a translation issue, and I wonder if you can find something that works better. You could try 'screening that is not recommended by local guidelines' or something similar. I also struggle with the comment about lay people's "common sense" (p 8, line 8-9). Perhaps either explain why you think it's common sense to attend screening more often than recommended, or use a different phrase.

Authors: we changed 'medically not indicated breast cancer screening' with 'screening that is not recommended by local guidelines' or 'without a medical reason/indication'. We also changed the

sentence about 'common sense' with a part on the social pressure and moral obligation in favour of breast cancer screening (*Moreover, social pressure in favour of breast cancer screening may stimulate a sense of moral obligation to participate*).

I struggled to understand your description of the studies on p 8, line 29-30. For example, even though you had briefly introduced the theory of regulatory fit, I didn't know what you meant by 'comparing two fit conditions vs. two non-fit conditions'. The meaning is explained later, but it's confusing in this section, I'd encourage you to explain more, or to leave it out of the introduction and put all the details in the methods section.

Authors: we have changed the full paragraph of the theory of regulatory fit. We added more details about the theory, the fit and non-fit conditions in the introduction. We added also many more details regarding the method of the two studies in a supplementary material file included in the present submission.

Methods:

As mentioned, I'd suggest you combine the methods sections.

Authors: thank you for your suggestion. Please see our previous answer regarding the same topic.

Results:

Some of the text here is quite dense with quantitative results and confidence intervals, making it very difficult to read. I'd suggest putting the quantitative detail in tables, and using the text just to highlight important findings, without repeating information that is in tables. Please note that I am not a quantitative researcher and cannot provide specialist statistical review.

Authors: we moved the quantitative results from text to Table. We now describe the main results in the text and all the statistical details are in the table.

Discussion:

I was interested in your inconsistent results between the two phases of the study. I wonder whether the nationality of the participants had an impact (study 1 was predominantly with Swiss nationals, study 2 was predominantly Italian nationals). Perhaps you could comment on this. For example, is there a different cultural milieu in the two groups that might make the population primed to receive or primed to ignore your intervention?

Authors: There might be a different cultural milieu between Swiss-Italian and Italian participants that might make the population primed to receive or primed to ignore the intervention. However, to the best of our knowledge there is no study comparing different cultural environments in the propensity to be primed or not. There are differences in the prevention vs. promotion regulatory focus orientation according to the culture. Individuals from collective cultures are more prone to have a prevention focus orientation, while individuals from individualistic cultures a promotion focus orientation. In this sense, it should be noticed that Italian and Swiss-Italian individuals share the same language and a very common cultural background. Moreover, the breast cancer prevention programs itself follow the same rules in Italy and Switzerland-Canton Ticino in term of age threshold and rules of inclusion. Anyway, we are grateful for your consideration, we added a sentence in the discussion about this topic.

Again, there was limited reference to the vast literature exploring barriers / enablers to adherence to breast cancer screening guidance. This might help to situate your study more clearly in the canon, assist with your interpretation and help to guide future research.

Authors: as reported above, we agree with you on this point. There is a vast amount of literature exploring barriers/facilitators to adherence to breast cancer screening guideline. Again, a complete review of this literature is beyond the scope of our paper. We have added a sentence and references in the introduction discussing this point and discussed it in the first paragraph of the discussion section.

Reviewer: 2

Reviewer Name: Jolyn Hersch

Institution and Country: The University of Sydney, Australia

Please state any competing interests or state 'None declared': None declared

This paper reports two connected studies seeking to address an important public health objective, namely reducing inappropriate overuse of screening (in this case mammography among women under 50 years of age). This research uses psychological theory to generate and test hypotheses about enhancing communication around this topic by taking into account motivational factors that may interact with information provision. I think this work could ultimately make an interesting and valuable contribution to the literature, and I commend the authors for their efforts. However, the research is somewhat complex and I feel the manuscript needs some refinement before it would be suitable for publication. Please see detailed comments below.

Authors: thank you for your appreciation and suggestions. We have revised the paper according to your comments and the other two reviewers' comments via track changes.

MAIN POINTS

1. Title: I find the title ambiguous with respect to the kind of 'adherence to recommendations' this paper addresses. The research purpose is articulated nicely at the top of p8: "to test whether health messages framed to correspond with a woman's regulatory focus orientation are effective in reducing the intention to ask for medically not indicated breast cancer screening under the age of 50". Could authors consider whether their title could be more informative by incorporating some of this wording (re reducing screening intentions in women under 50)?

Authors: thank you for your suggestion. The first reviewer found the expression 'medically not indicated breast cancer screening' confusing and ask to explain better what we intend to say. We have considered including a similar expression in the title, but it might be confusing as suggested by the first review. Therefore, even if we appreciate your comment and suggestion, we decide to keep the original title

2. Abstract: Could the authors consider including some wording (e.g. 'overuse' or a related term) in the abstract to facilitate connecting the paper with the broader literature around the topic?

Authors: thank you for your suggestion; please see the new version of the abstract

3. I think table 2 should be converted into two flow charts to show more clearly the sequence of steps in each study, including randomisation to conditions, measurement of variables at time points, manipulation/priming of orientation, tailoring of messages to personal orientation etc.

Authors: we have created two flowcharts explaining the sequence of steps for each study. Thank you for the suggestion, very helpful!

4. As intention is the primary outcome, more detail is needed about its measurement (p11). What was the response scale/options and possible range of scores? This is crucial to interpret table 4.

Authors: we have created a detailed supplementary material with all the measures description, included the items measuring the intention.

5. Analytic strategy (study 1, p11) needs elaboration – for example ‘We used analysis of covariance to...’ Also, I notice that “There were not missing data.” Is that because the survey required women to answer every question before progressing to the next screen? If so, please state this.

Authors: for both study 1 and study 2, we have added explanation regarding the aims of the analyses performed. We have also specified that absence of missing data was due by the fact women have to answer to every question before progressing, as you mentioned.

6. Results, opening sentence (study 1, p11): Please double-check the numbers – is it just coincidence that the numbers are exactly the same for both conditions?

Authors: it is just a coincidence; the results are exactly as we have reported in the text

7. Discussion (study 1, p12): Please include a sentence here to briefly recap the main finding/s.

Authors: discussion of study 1 has been deleted as required by the Editorial Team. We have created a unique description of the findings for study 1 and study 2 in the final discussion section.

8. “292 participants were randomly assigned to five conditions” (p13) – wouldn’t it be accurate to say 973 women in the pre-test sample were randomised? Also, does that sentence contradict this? “Participants were randomly assigned to the fit or non-fit condition or the control group.”

Authors: actually it would not. 973 women have started the first part of the survey; 292 (out of 973) have started and finished the first and the second part of the survey.

The 681 participants who have started the first part did not give us the possibility to be contacted for the second part of the research, because they did not give us their emails. The randomization was done with the women who finished the first survey and sent us their emails. Those women were randomized and contacted for the second part. Therefore, the 681 were not randomized into the five conditions.

9. Procedure (study 2, p14): “...questionnaire measuring women's regulatory focus. The latest was applied because working with the trait regulatory focus would be more stable than a primed focus...” It seems to me this is an important point about a key difference between studies 1 and 2. I suggest explaining this in the discussion section of study 1 to help readers understand why study 2 was done (e.g. to address a limitation of study 1, explore a different aspect of theory).

Authors: yes, we agree with your consideration. As the Editorial Team asked to delete the discussion of the Study 1, we have discussed your suggested point in the final discussion section

10. I’m confused by these two sentences (p15) as they appear similar yet different: “The within-subject effect reached significance, $F(1, 268.25)=5.34$, $p=.021$, with a general decreasing of the intention from the pre-test ($M=3.38$, $SD=1.32$) to the post-test measurement ($M=3$, $SD=1.52$).” /

“There was a significant within subjects effect, $F(1, 267.91)=5.10$, $p=.025$, partial $\eta^2=.02$, indicating that there was a general decrease of the intention from pre to post-evaluation...”

Authors: we have deleted the sentence “The within-subject effect reached significance, $F(1, 268.25)=5.34$, $p=.021$, with a general decreasing of the intention from the pre-test ($M=3.38$, $SD=1.32$) to the post-test measurement ($M=3$, $SD=1.52$.” to avoid any confusion

11. Several times throughout the manuscript the authors appear to cite their own previous work as ‘blind for review’. Perhaps this was necessary for previous submission to a different journal, but I don’t think it’s required for BMJ Open since peer review is fully open anyway. When submitting any revision of this manuscript please include full details of all references.

Authors: ye, you are right. It is a typo. We have replaced the blind for reviews

MINOR POINTS

12. Introduction (p6): “possible harms associated with regular mammography below the age of 50 are recognized” – please name the main harms (i.e. overdiagnosis and false positives).

Authors: thank you, done

13. Introduction (p6): “many women below the age of 50 seek and receive screenings” – I haven’t checked the references but assume they cover data from various countries? I suggest saying so.

Authors: yes, you are right. done

14. This is a matter of taste and the authors might disagree, but I suggest moving the sentences about recruitment and eligibility (bottom of p9) to earlier, under the heading Participants. (And likewise for study 2). Also, although the Facebook advertisements obviously were targeted to women in Ticino, they presumably appeared on some public Facebook pages – so was living in Ticino actually part of the eligibility criteria or could anyone visiting these pages take part?

Authors: the reference to the Ticino pages was deleted because redundant. We advertised the post on the Facebook page of the University. We also moved the recruitment and eligibility part under the heading Participants.

15. I have never heard of the Personal Involvement Inventory (p10) – could the authors please add a sentence to briefly explain what it is and why it was included?

Authors: the Personal Involvement Inventory is a brief questionnaire measuring the ego involvement of an individual into a specific topic, breast cancer in our case. Some of the authors of the present research have applied this questionnaire in a preliminary study on Swiss women (Labrie et al, 2020). They have found that the level of the ego involvement influences the women’s intention to ask for a breast cancer screening. That is the reason why we include this construct in the present research. You may find a detailed description of this measure into the supplemental material we have created.

16. On p13 “Italian and Ticinese-Swiss participants ... follow the same rules for their breast cancer screening programs.” Please briefly state the program ‘rules’ referred to (here or in the intro).

Authors: done, please find the explanation on page 12

17. I am not sure what you mean by “the 'just-feels-right' experience” (p16) – please explain.

Authors: we added this part to the introduction: *If a person performs a behaviour or processes a message highlighting goal pursuit strategies that match their regulatory orientation, they experience a phenomenon called “regulatory fit” (Higgins, 2000). For example, if a person with a promotion orientation reads a message highlighting strategies to achieve gains, a fit condition occurs. The same applies to someone with a prevention orientation processing a message emphasizing strategies to avoid losses. Such a fit or match causes an “it just feels right” perception increasing the perceived value of the behaviour (Cesario, Grant, & Higgins, 2004).* We expanded the introduction section and better explained the theory of regulatory fit

18. Please briefly describe the process of obtaining informed consent from study participants.

Authors: Before starting the questionnaire, participants have to read the information sheet and declare if they want to participate or not, by clicking on the corresponding button (i.e., “yes, I want to participate”; “no, I do not want to participate”). We added info about this, please see page 8 and 11

Reviewer: 3

Reviewer Name: Stephen W. Duffy

Institution and Country: Queen Mary University of London, UK

Please state any competing interests or state ‘None declared’: None

1. This paper reports on a comparison of different communication strategies aimed at changing health behaviour with respect to mammography screening participation. The study design seems appropriate and the work has been conducted correctly, but two major problems with this paper occur to the reader. First, it seems to be addressing the wrong problem. In Europe and North America, the problem isn't too many people seeking mammography screening: the problem is that large numbers of women who need mammography screening do not get it. In the UK, 30% of those invited to the national programme do not attend. The figure is higher in many other European countries.

Authors: we agree with your consideration that there is an amount of women who do not attend mammography screening when they are invited. There are several risk conditions that should make women to attend the required breast cancer screening, such as for example BRCA 1 or BRCA2 mutation, or history of breast cancer among relatives. In addition, even in case of no-risk conditions, a large amount of women *invited* to the national programme do not attend.

However, our research focuses on young women, under the age of 45, without any risk conditions. Our aim was to inform them about the balance *for their age group* between the benefits of the breast cancer screening and the associated harms. Those women *are not invited* to attend the national programme because they are young and without risk conditions. In absence of risk conditions, the age threshold to start inviting women is still under debate (Bucchi et al., 2019; Gøtzsche & Jørgensen, 2013; van den Ende, Oordt-Speets, Vrolijk & van Agt, 2017). We know that for young women the balance between the benefits (i.e., reducing breast cancer mortality) and the harms associated with mammography (i.e., x-ray, over diagnosis, false positive results) is uncertain. For all these reasons recommended age for starting with the screening varied from 40 (Mainiero et al., 2016), to 45 (ECIBC Guideline, 2018; Oeffinger et al., 2015), to 50 years (Lauby-Secretan et al., 2015; Siu, 2016).

The present research was conducted in Italy and Switzerland, two countries that follow the European Guidelines (ECIBC Guideline, 2018) stating that women under 45, with no risk conditions, should not have systematic breast cancer screenings. However, we found that many women below the

established age threshold seek and receive mammography screenings without medical reasons in the U.S. (Block, Jarlenski, Wu, & Bennett, 2013; Kapp, Reyerson, Couchlin, & Thompson, 2009) Switzerland (Glaus, Fäh, Hornung, Senn, & Stiefel, 2004; Labrie, Ludolph, & Schulz, 2020), Germany (Klug, Hetzer, & Blettner, 2005), and The Netherlands (Statistics Netherlands, 2015).

This is a complementary problem compared to the one of women invited and not attending the screening. In both cases, researchers should test which are the best communication strategies in order to promote informed decision-making, in line with the more recent medical guidelines, to promote individual's health.

2. A second issue is the assumption that screening in the general population in women aged under 50 is assumed to be detrimental. The paper justifies this by reference to a highly selective set of publications. There is evidence from both experimental (e.g., Nyström et al, J Med Screen 2017; 24: 34-42) and observational studies (e.g, Hellquist et al, Cancer 2011; 117: 714-22) that mammography screening in women aged 40-49 is effective in preventing breast cancer deaths. Some balance here would be welcome.

Authors: we revised the first part of the introduction, we added up-to-date references to the most recent guidelines as follow: *Breast cancer is one of the most common forms of cancer in women worldwide and the principal cause of cancer-related death in the female population (Torre, Islami, Siegel, Ward, & Jemal, 2017). To promote early diagnosis, many EU countries have introduced systematic breast cancer screening programs (Altobelli & Lattanzi, 2014). However, the age threshold to start inviting women to those screening is under debate (Bucchi et al., 2019; Gøtzsche & Jørgensen, 2013; van den Ende, Oordt-Speets, Vroling & van Agt, 2017). The balance between the benefits, that is reducing breast cancer mortality, and the harms associated with mammography, such as over diagnosis and false positive results (Armstrong, Moye, Williams, Berlin, & Reynolds, 2007; Barratt, Howard, Irwig, Salked, & Houssami, 2005; Gøtzsche & Nielsen, 2006; Gøtzsche & Jørgensen, 2013; van den Ende, Oordt-Speets, Vroling & van Agt, 2017) is uncertain. Technologies for breast cancer screening evolve and the changing affect evidence quality and suggested recommendations (Carte et al., 2015). For these reasons recommended age for starting with the screening varied from 40 (Mainiero et al., 2016), to 45 (ECIBC Guideline, 2018; Oeffinger et al., 2015), to 50 years (Lauby-Secretan et al., 2015; Siu, 2016).*

3. The reader is somewhat thrown in at the deep end with respect to the language of social psychology, from abstract onwards. I am not sure what could be done about this without unacceptable increases in word counts. However, I suggest that the authors try to signpost the concepts for the more general reader.

Authors: we have considerably reviewed the abstract and the introduction section, especially the parts about the regulatory fit and focus orientation avoiding any jargon related to social psychology. We saved words cutting the method section and creating a supplementary material, which is included in the present re-submission

4. It is a limitation that post-intervention, only intention is reported, not actually screening participation. Is it possible to obtain data on actual participation post-intervention? If not, this should be acknowledged as a limitation.

Authors: you are right; it would be great to have the chance to collect data on actual behaviors. Unfortunately, the timeline of the research project and the resources we had did not allow us to add this measurement point to study 1 or study 2. Anyway, according to many theories in health promotion (see for example, the Health Belief Model), the intention is considered a good predictor of the actual

behavior. However, we understand your consideration and we added a sentence on the limitation section.

VERSION 2 – REVIEW

REVIEWER	Lisa Parker University of Sydney, Australia
REVIEW RETURNED	23-Jul-2020

GENERAL COMMENTS	I think this is an important study and I look forward to seeing it in print. As with my previous comments, I recommend you write this up as a single study, and provide a single Methods section and a single Results section. You can describe the two sub-studies as linked phases of the same study, or similar terminology Discussion – I suggest leading this section with a brief summary of your results, e.g. move the current paragraph 3 up to become para 1. Possibly paragraphs 1 and 2 belong in the introduction anyway? Some minor points P3 line 5 – I suggest changing ‘prevent’ to ‘reduce’ P5 line 32 – the age is not ‘established’ but rather is simply ‘recommended’ – as you say in your previous paragraph P7 line 18 – I suggest avoiding the phrase ‘non at risk’ women, all women are at risk of having breast cancer. Maybe substitute with ‘non high-risk’ women There is some slippage between ‘mammography screening without medical indication’ and ‘mammography without medical indication’ (e.g. p 3, line 46). These are very different and I am sure you are not recommending that women with symptoms should avoid diagnostic mammography. I still struggle with the phrase ‘mammography screening without medical indication’ – in my mind, the term ‘medical indication’ refers to symptoms, and screening is, by definition, a test for people without symptoms; i.e. all breast screening is for women without medical indication, instead the indication is a public health one. Could you use something like ‘mammography screening outside the recommended guidelines’?
--

REVIEWER	Dr Jolyn Hersch The University of Sydney, Australia
REVIEW RETURNED	21-Jul-2020

GENERAL COMMENTS	An Application of the Theory of Regulatory Fit to Promote Adherence to Evidence-Based Breast Cancer Screening Recommendations: Experimental vs. Longitudinal Evidence Journal: BMJ Open
--

Manuscript: bmjopen-2020-037748.R1

The authors have substantially revised their manuscript and addressed many of my original points. However, I still have some important concerns pertinent to the revised manuscript, outlined below.

1. Abstract

a. I appreciate that the authors have now followed my suggestion and included some wording around 'overuse' in the abstract.

However, the newly added opening phrase, "To prevent overtreatment caused by false positive results and overuse" does not quite work because overtreatment is caused by overdiagnosis, not by false positive results. I suggest amending this, perhaps simply to "prevent overtreatment caused by overuse of screening".

b. The sentence "Fear of breast cancer and risk perception are significantly related to intention to seek mammography" should be in the past tense ("were" instead of "are") because it reflects the results of this specific study rather than a broader generalisation.

2. Study 1 methods

a. The manuscript says "Regulatory priming manipulation was induced... Participants were then randomly assigned" but surely it should be the other way around (as shown nicely in Figure 1)? Randomisation first, followed by priming?

b. I appreciate that the authors have provided detailed information about the measurement of intention in the supplementary material. However, I feel strongly that the primary outcome must be described properly in the manuscript, including the possible range of final scores and the meaning of higher versus lower scores. This is essential for readers to be able to interpret the numbers in table 2 and understand which numbers represent high intentions etc.

3. Study 2 methods

a. I still find this sentence problematic: "292 participants were randomly assigned to 5 conditions" (under the heading Participants). I think it could be left out; the process is more clearly outlined in a subsequent paragraph: "the regulatory focus orientation was measured... Subsequently, participants were randomly assigned" to the various conditions. Presumably participants were first stratified into prevention and promotion orientation groups (based on the questionnaire), and then randomised within those groups? In other words, randomisation was performed separately for prevention-oriented women and promotion-oriented women to ensure a balanced representation of orientations between the match and non-match conditions – right? Please clarify.

b. Figure 2 should be amended to clarify the process too, taking into account the above comment.

c. Why was the control group so small?

4. Study 2 methods (supplementary material)

a. Firstly, please clarify how each individual's chronic orientation was calculated instead of just saying "following the original procedure" with a reference that the reader then has to look up. Now that the methods are in supplementary material, extra words are less of a concern. Were participants classified using a median split on the difference between their RFQ Promotion and RFQ Prevention scores, or in another way?

	b. Secondly, the heading Experimental Manipulation applies only to the Video Messages in this study, so it should be moved to after the paragraph about Trait Regulatory Orientation. c. Thank you for adding explanation about the Personal Involvement Inventory (supplementary material) but there is a sentence with missing words that needs to be fixed: "...measuring participants' involvement in breast cancer screening through affective and cognitive adjectives because previous research has found that . The scale was administered..." 5. Finally, please use "non-fit" throughout instead of "unfit".
--	---

REVIEWER	Stephen W. Duffy Queen Mary University of London, UK
REVIEW RETURNED	20-Jul-2020

GENERAL COMMENTS	The authors have done their best to respond to the reviewers and only a few issues remain:  1. The abstract still seems rather jargon-loaded. 2. In the abstract, there seems to be confusion between overdiagnosis and false positives. False positives are persons recalled for assessment of a suspicious finding who turn out not to have cancer. Overdiagnosis is the diagnosis of cancer, histologically confirmed, which would not have been diagnosed in the host's lifetime had screening not taken place. One might have overinvestigation due to false positives, but overtreatment is a consequence of overdiagnosis, not of false positives. 3. There remains a lack of balance in terms of the literature referred to (see my previous comment 2).
--

VERSION 2 – AUTHOR RESPONSE

Reviewer(s)' Comments to Author:

Reviewer Name

**

Stephen W. Duffy

Institution and Country

Queen Mary University of London, UK

Please state any competing interests or state 'None declared':

None

The authors have done their best to respond to the reviewers and only a few issues remain:

1. The abstract still seems rather jargon-loaded.

Authors: we deleted all the jargon related to the theory of regulatory fit. We deleted any reference to the goal-pursuit orientation and we referred to them with the more basic word "motivation orientation". We deleted any reference to the to the promotion focus and prevention focus orientation and we explain in plain English what the two orientations mean.

2. In the abstract, there seems to be confusion between overdiagnosis and false positives. False positives are persons recalled for assessment of a suspicious finding who turn out not to have cancer. Overdiagnosis is the diagnosis of cancer, histologically confirmed, which would not have been diagnosed in the host's lifetime had screening not taken place. One might have overinvestigation due to false positives, but overtreatment is a consequence of overdiagnosis, not of false positives.

Authors: yes, you are right. We corrected the sentence

3. There remains a lack of balance in terms of the literature referred to (see my previous comment 2).

Authors: many thanks, we added a full paragraph and few spare lines in the introduction.

**

Reviewer Name

Dr Jolyn Hersch

Institution and Country

The University of Sydney, Australia

Please state any competing interests or state 'None declared':

None declared

Please leave your comments for the authors below

An Application of the Theory of Regulatory Fit to Promote Adherence to Evidence-Based Breast Cancer Screening Recommendations:

Experimental vs. Longitudinal Evidence

Journal: BMJ Open

Manuscript: bmjopen-2020-037748.R1

The authors have substantially revised their manuscript and addressed many of my original points. However, I still have some important concerns pertinent to the revised manuscript, outlined below.

1. Abstract

a. I appreciate that the authors have now followed my suggestion and included some wording around 'overuse' in the abstract. However, the newly added opening phrase, "To prevent overtreatment caused by false positive results and overuse" does not quite work because overtreatment is caused by overdiagnosis, not by false positive results. I suggest amending this, perhaps simply to "prevent overtreatment caused by overuse of screening".

Authors: you are right. We have corrected the sentence in the abstract

b. The sentence "Fear of breast cancer and risk perception are significantly related to intention to seek mammography" should be in the past tense ("were" instead of "are") because it reflects the results of this specific study rather than a broader generalisation.

Authors: done

2. Study 1 methods

a. The manuscript says “Regulatory priming manipulation was induced... Participants were then randomly assigned” but surely it should be the other way around (as shown nicely in Figure 1)? Randomisation first, followed by priming?

Authors: yes, many thanks, you are right. We have changed the sentence in the method section

b. I appreciate that the authors have provided detailed information about the measurement of intention in the supplementary material. However, I feel strongly that the primary outcome must be described properly in the manuscript, including the possible range of final scores and the meaning of higher versus lower scores. This is essential for readers to be able to interpret the numbers in table 2 and understand which numbers represent high intentions etc.

Authors: we included information about range of the final scores and meaning of higher scores for both study 1 and study 2 (see supplementary material)

3. Study 2 methods

a. I still find this sentence problematic: “292 participants were randomly assigned to 5 conditions” (under the heading Participants). I think it could be left out; the process is more clearly outlined in a subsequent paragraph: “the regulatory focus orientation was measured... Subsequently, participants were randomly assigned” to the various conditions. Presumably participants were first stratified into prevention and promotion orientation groups (based on the questionnaire), and then randomised within those groups? In other words, randomisation was performed separately for prevention-oriented women and promotion-oriented women to ensure a balanced representation of orientations between the match and non-match conditions – right? Please clarify.

Authors: you are right. We have changed the sentence starting with “292 participants were randomly...”. Yes, participants were stratified first and then assigned to one group. We specified this point in the Process, Measures and Data Collection section.

b. Figure 2 should be amended to clarify the process too, taking into account the above comment.

Authors: we have changed figure 2 as you suggested

c. Why was the control group so small?

Authors: unfortunately there was a high dropout rate, we do not have any particular explanation for that

4. Study 2 methods (supplementary material)

a. Firstly, please clarify how each individual's chronic orientation was calculated instead of just saying "following the original procedure" with a

reference that the reader then has to look up. Now that the methods are in supplementary material, extra words are less of a concern. Were participants classified using a median split on the difference between their RFQ Promotion and RFQ Prevention scores, or in another way?

Authors: we have added information regarding the procedure. Higgins suggested to subtract RFQ Promotion to RFQ Prevention.

b. Secondly, the heading Experimental Manipulation applies only to the Video Messages in this study, so it should be moved to after the paragraph about Trait Regulatory Orientation.

Authors: done, many thanks

c. Thank you for adding explanation about the Personal Involvement Inventory (supplementary material) but there is a sentence with missing words that needs to be fixed: "...measuring participants' involvement in breast cancer screening through affective and cognitive adjectives because previous research has found that . The scale was administered..."

Authors: thank you, we have corrected the sentence and added a ref

5. Finally, please use "non-fit" throughout instead of "unfit".

Authors: done

**

Reviewer Name

Lisa Parker

Institution and Country

University of Sydney, Australia

Please state any competing interests or state 'None declared':

none declared

Please leave your comments for the authors below

I think this is an important study and I look forward to seeing it in print.

As with my previous comments, I recommend you write this up as a single study, and provide a single Methods section and a single Results section. You can describe the two sub-studies as linked phases of the same study, or similar terminology

Authors: thank you for your suggestion. We have changed the organization of the paper as you suggested.

Discussion – I suggest leading this section with a brief summary of your results, e.g. move the current paragraph 3 up to become para 1. Possibly paragraphs 1 and 2 belong in the introduction anyway?

Authors: we have moved para 1 and 2 of the discussion in the introduction. The discussion now started with (old) para 3. Many thanks for your revisions

Some minor points

P3 line 5 – I suggest changing 'prevent' to 'reduce'

Authors: done

P5 line 32 – the age is not 'established' but rather is simply 'recommended' – as you say in your previous paragraph

Authors: done

P7 line 18 – I suggest avoiding the phrase ‘non at risk’ women, all women are at risk of having breast cancer. Maybe substitute with ‘non high-risk’ women

Authors: done

There is some slippage between ‘mammography screening without medical indication’ and ‘mammography without medical indication’ (e.g. p 3, line 46). These are very different and I am sure you are not recommending that women with symptoms should avoid diagnostic mammography.

I still struggle with the phrase ‘mammography screening without medical indication’ – in my mind, the term ‘medical indication’ refers to symptoms, and screening is, by definition, a test for people without symptoms; i.e. all breast screening is for women without medical indication, instead the indication is a public health one. Could you use something like ‘mammography screening outside the recommended guidelines’?

Authors: we changed ‘mammography screening without medical indication’ with ‘mammography screening outside the recommended guidelines’. Thanks for the suggestion

VERSION 3 – REVIEW

REVIEWER	Lisa Parker The University of Sydney, Australia
REVIEW RETURNED	24-Sep-2020

GENERAL COMMENTS	This is great work, the authors have done a good job of revising and the paper is much clearer. I have a couple of very minor comments: Abstract, Participants, you write: “360 women participated in Phase 1 and other 292 in Phase 2.” Perhaps you mean ‘another 292’? Into p 6, you write: “No previous studies have tested messages designed according to the assumptions of regulatory fit to influence the intention to not engage in disease detection screening. This would challenge the intuitive perception that breast cancer screening leads to a mortality reduction determined by breast cancer” I wonder whether you should qualify this last comment – do you mean breast cancer screening at any time, including outside the recommended guidelines or breast cancer screening within the guidelines? Discussion, paragraph 1, last sentence – you write that ‘the just-feels-right experience appears to be insufficient to convince non high-risk women under that age threshold to avoid systematic breast cancer screening in the long run.’ I am not sure why you include the final phrase, ‘in the long run’. It appears that your study
---

	is concentrating on young women who attend screening outside of the recommended program, i.e. at a younger age than recommended. Using 'in the long run' would imply that you are suggesting young women who are not at high risk should consider never attending screening, even once they reach the recommended age. If this is what you are suggesting then perhaps it should be explained in more detail. Otherwise, perhaps replace 'in the long run' with something like 'outside of the recommended guidelines.'
--	---

REVIEWER	Dr Jolyn Hersch The University of Sydney, Australia
REVIEW RETURNED	25-Sep-2020

GENERAL COMMENTS	Overall the authors have done a good job of addressing reviewer comments.
---

REVIEWER	Stephen W. Duffy Queen Mary University of London, UK
REVIEW RETURNED	28-Sep-2020

GENERAL COMMENTS	The authors have responded to previous concerns, and I have only one further comment. The statement in the introduction that 'The balance between the benefits (i.e., reducing breast cancer mortality) and the harm associated with mammography (i.e. x-ray exposure, over diagnosis and false positive results; see,[4-8]) is uncertain' is overdoing it. For women aged 50 years and above, almost all scientific and public health bodies agree that the balance is in favour of benefit. I suggest the sentence be rephrased as: 'The balance of benefits and harms of mammography screening is less certain women aged under 50'.
---